# NMR-Based Metabolomic Approach Tracks Potential Serum Biomarkers of Disease Progression in Patients with Type 2 Diabetes Mellitus

**DOI:** 10.3390/jcm8050720

**Published:** 2019-05-21

**Authors:** Laura Del Coco, Daniele Vergara, Serena De Matteis, Emanuela Mensà, Jacopo Sabbatinelli, Francesco Prattichizzo, Anna Rita Bonfigli, Gianluca Storci, Sara Bravaccini, Francesca Pirini, Andrea Ragusa, Andrea Casadei-Gardini, Massimiliano Bonafè, Michele Maffia, Francesco Paolo Fanizzi, Fabiola Olivieri, Anna Maria Giudetti

**Affiliations:** 1Department of Biological and Environmental Sciences and Technologies, University of Salento, via Monteroni, 73100 Lecce, Italy; laura.delcoco@unisalento.it (L.D.C.); daniele.vergara@unisalento.it (D.V.); andrea.ragusa@unisalento.it (A.R.); michele.maffia@unisalento.it (M.M.); anna.giudetti@unisalento.it (A.M.G.); 2Biosciences Laboratory, Istituto Scientifico Romagnolo per lo Studio e la Cura dei Tumori (IRST) IRCCS, 47014 Meldola, Italy; serena.dematteis@irst.emr.it (S.D.M.); sara.bravaccini@irst.emr.it (S.B.); francesca.pirini@irst.emr.it (F.P.); 3Department of Clinical and Molecular Sciences, DISCLIMO, Università Politecnica delle Marche, 60121 Ancona, Italy; emanuela.mensa@gmail.com (E.M.); jacopo.sabbatinelli@gmail.com (J.S.); 4IRCCS MultiMedica, 20099 Milan, Italy; francesco.prattichizzo@multimedica.it; 5Scientific Direction, IRCCS INRCA, 60121 Ancona, Italy; a.bonfigli@inrca.it; 6Department of Experimental, Diagnostic and Specialty Medicine, University of Bologna, 40126 Bologna, Italy; gianluca.storci@unibo.it (G.S.); massimiliano.bonafe@unibo.it (M.B.); 7Division of Medical Oncology, Department of Medical and Surgical Sciences for Children and Adults, University Hospital of Modena, 41125 Modena, Italy; casadeigardini@gmail.com; 8Center of Clinical Pathology and Innovative Therapy, IRCCS INRCA, 60121 Ancona, Italy

**Keywords:** branched-chain amino acids, metabolomics, NMR spectroscopy, type 2 diabetes mellitus

## Abstract

Type 2 diabetes mellitus (T2DM) is a metabolic disorder characterized by chronic hyperglycemia associated with alterations in carbohydrate, lipid, and protein metabolism. The prognosis of T2DM patients is highly dependent on the development of complications, and therefore the identification of biomarkers of T2DM progression, with minimally invasive techniques, is a huge need. In the present study, we applied a ^1^H-Nuclear Magnetic Resonance (^1^H-NMR)-based metabolomic approach coupled with multivariate data analysis to identify serum metabolite profiles associated with T2DM development and progression. To perform this, we compared the serum metabolome of non-diabetic subjects, treatment-naïve non-complicated T2DM patients, and T2DM patients with complications in insulin monotherapy. Our analysis revealed a significant reduction of alanine, glutamine, glutamate, leucine, lysine, methionine, tyrosine, and phenylalanine in T2DM patients with respect to non-diabetic subjects. Moreover, isoleucine, leucine, lysine, tyrosine, and valine levels distinguished complicated patients from patients without complications. Overall, the metabolic pathway analysis suggested that branched-chain amino acid (BCAA) metabolism is significantly compromised in T2DM patients with complications, while perturbation in the metabolism of gluconeogenic amino acids other than BCAAs characterizes both early and advanced T2DM stages. In conclusion, we identified a metabolic serum signature associated with T2DM stages. These data could be integrated with clinical characteristics to build a composite T2DM/complications risk score to be validated in a prospective cohort.

## 1. Introduction

Diabetes mellitus (DM) is a metabolic disorder characterized by chronic hyperglycemia associated with impairments in carbohydrate, lipid, and protein metabolism [1]. DM is classified in two main categories: type 1, due to cellular-mediated autoimmune pancreatic islet β-cells destruction which occurs in 5–10% of cases, and type 2 (T2DM), due to insulin resistance (IR) with a defect in compensatory insulin secretion, which affects 90% of diabetic patients [2]. Environmental and lifestyle changes in association with populations aging account for the rapid global increase in T2DM prevalence and incidence in recent decades [3]. A comprehensive summary of factors contributing to T2DM risk includes not only obesity and related aspects of diet quality and quantity, but also sedentary lifestyle and lack of physical activity, exposure to noise or fine dust, short or disturbed sleep, smoking, stress, depression, and a low socioeconomic status [4]. Chronic hyperglycemia leads to many long-term complications, such as cardiovascular disease (CVD), cerebrovascular disease, peripheral vascular disease, neuropathy, retinopathy, and renal failure, resulting in increasing disability, reduced life expectancy, and increased health costs [5]. The prognosis of patients with T2DM is highly dependent on the development of complications; the prevalence of patients with cardiovascular complication is growing exponentially and most T2DM patients die as a result of cardiovascular causes. 

Despite the recently introduced therapies to manage T2DM, the progressive nature of IR and the inability of β-cells to cope with increased insulin demand still obligate insulin therapy for selected clusters of patients to achieve and maintain adequate glycemic control [6,7,8]. 

Because of the risks and benefits of treatments, the care of patients with T2DM involves complex decision-making [9]. Therefore, the identification of biomarkers of metabolic worsening in T2DM is of clinical relevance.

Recently, several metabolomic techniques were applied to the identification of metabolic signatures associated with T2DM [10,11,12,13,14,15,16,17]. Data in this research field were extensive, revealing new diagnostic/prognostic disease biomarkers [14], or metabolic markers of response to targeted therapies [15,16]. Metabolomics also provided tools for patient’s stratification and recognizable metabolic patterns associated with organ dysfunction [17]. These studies focused on the diagnostic value of metabolic signatures in prospective cohorts, while the metabolic comparison of T2DM patients at different stages of the disease was less investigated. To do this, we took advantage of a selected cohort of T2DM patients that we characterized in our previous works [18,19,20,21,22,23]. 

Here, we aimed to identify, by a ^1^H-Nuclear Magnetic Resonance (^1^H-NMR)-based metabolomic approach coupled with multivariate data analysis, a serum metabolic signature associated with T2DM development and progression. The study design is characterized by the comparison of opposite phenotypes: non-diabetic subjects and two subsets of T2DM patients—T2DM patients at an early stage of the disease (without complications and treatments) and at a late stage (with diabetic complications on insulin monotherapy).

## 2. Materials and Methods

### 2.1. Patient Samples

Enrolled patients were accurately selected from a large cohort of Italian T2DM patients and control subjects, recruited from the Italian National Research Center on Aging (INRCA), Ancona. All subjects provided written informed consent, which was approved by the INRCA’s Ethics Committee. The inclusion criteria for T2DM patients and the clinical information collected from each subject were as described by Testa et al. [24].

We selected 26 T2DM patients and seven control subjects with comparable age, body mass index (BMI), lipid plasma profile, and gender distribution. All studied subjects consumed a Mediterranean diet. Subjects were considered as controls if at the time of blood collection they did not have T2DM and any major acute and/or chronic age-related diseases such as acute myocardial infarction, chronic heart failure, Alzheimer’s disease, or cancer. Among the 26 T2DM patients, we selected 13 patients without complications and 13 patients with documented complications on insulin monotherapy at the time of blood collection. The presence/absence of diabetic complications was established as follows:(1)retinopathy was defined as dilated pupils detected on funduscopic and/or fluorescence angiography;(2)incipient nephropathy was a urinary albumin excretion rate >30 mg/24 h and normal creatinine clearance;(3)chronic renal failure was defined as an estimated glomerular filtration rate <60 mL/min per 1.73 m^2^, based upon the four-variable modification of diet in renal disease (MDRD) formula;(4)neuropathy was established by electromyography;(5)ischemic heart disease was diagnosed by clinical history and/or ischemic electrocardiographic alterations; these patients had had ST- or non-ST-elevation myocardial infarction, which was defined as a major adverse cardiac event (MACE);(6)Peripheral vascular disease, including arteriosclerosis obliterans and cerebrovascular disease, was diagnosed based on history, physical examination, and Doppler imaging.

Each patient could be affected by more than one complication (Table 1). 

Uncomplicated patients were recruited within one month of diagnosis and had not received any specific pharmacological treatment for diabetes at the time of the blood collection. Other drugs prescribed to diabetic patients are shown in the Table 2. To avoid possible bias due to different treatments, T2DM patients taking glucose-lowering drugs others from insulin were excluded.

### 2.2. Laboratory Assays

Overnight fasting venous blood samples were collected from 08:00 to 10:00 h. Blood concentration of glycosylated hemoglobin (HbA1c) was measured by a G8 HPLC analyzer (TOSOH BIOSCIENCE, Tokyo, Japan). Plasminogen activator inhibitor-1 (PAI-1) antigen was quantified with an immune-enzymatic method (Biopool, Sweden). Total and high-density lipoprotein (HDL) cholesterol, triacylglycerols (TAG), fasting insulin, fasting glucose, fibrinogen, apolipoprotein AI and B (ApoAI and ApoB), and creatinine were measured using commercially available kits on an automated clinical chemistry COBAS Modular Platform (C module) analyzer (Roche-Hitachi, Basel, Switzerland). Highly sensitive C-reactive protein (hsCRP) was determined by the particle-enhanced immune-turbidimetric assay (CRP High Sensitive, Roche-Hitachi) on a COBAS analyzer. The homeostasis model assessment (HOMA) index was calculated as glucose (mg/100 mL)*insulin (uIU/mL)/405. BMI was calculated as weight divided by height squared (kg/m^2^). α-Fucosidase and β-galactosidase were quantified as described by Spazzafumo et al. [25]. Insulin growth factor 1 (IGF1) level was quantified with a commercially available ELISA kit.

### 2.3. Sample Preparation and NMR Measurements

Serum samples were stored at a temperature of −80 °C until the NMR measurements were performed. Prior to NMR analysis, serum samples were thawed and an aliquot of 200 μL was mixed with 400 μL saline buffer solution (in 100% D2O containing TSP as a chemical shift reference, δ = 0 ppm, NaCl 0.9%, 50 mM sodium phosphate buffer and pH 7.4) to minimize the pH variation and transferred in a 5 mm NMR tube [26]. All measurements were performed on a Bruker Avance III 600 Ascend NMR spectrometer (Bruker, Ettlingen, Germany), operating at 600.13 MHz for 1H observation, equipped with a TCI cryoprobe (Triple Resonance inverse Cryoprobe) incorporating a z-axis gradient coil and automatic tuning-matching (ATM). Experiments were acquired at 300 K in automation mode after loading individual samples on a Bruker Automatic Sample Changer, interfaced with the IconNMR software (Bruker). For each sample, two types of 1D ^1^H-NMR experiments were recorded: a standard (ZGCPPR Bruker standard pulse sequence) spectrum, with pre-saturation and composite pulse for selection, and a Carr–Purcell–Meiboom–Gill (CMPG) spin-echo sequence, with 32 transients, 16 dummy scans, 5 s relaxation delay, size of FID (free induction decay) of 64 K data points, spectral width of 12,019.230 Hz (20.0276 ppm), an acquisition time of 1.36 s, a total spin-spin relaxation delay of 1.2 ms, and solvent signal saturation during the relaxation delay. The resulting FIDs were multiplied by an exponential weighting function corresponding to a line broadening of 0.3 Hz before Fourier transformation, automated phasing, and baseline correction. Moreover, peak assignments were carried out using 2D NMR experiments (^1^H-^1^H J-resolved, ^1^H-^1^H COSY, Correlation Spectroscopy, ^1^H-^13^C HSQC Heteronuclear Single Quantum Correlation, ^1^H-^13^C HMBC, Heteronuclear Multiple Bond Correlation) and by comparison with published data [26,27].

### 2.4. Metabolic Pathway Analysis

The most relevant metabolic pathways potentially involved in the metabolomic study were identified using MeTPA MetaboAnalyst software [28]. The purpose is to investigate if certain metabolic pathways are significantly different for the two groups of patients, when compared with control subjects and also whit each other. Metabolites of interest previously quantified by selected distinctive unbiased NMR signals were used as the input matrix for the metabolic pathway analysis. The pathway impact is calculated as the sum of the importance measures of the matched metabolites normalized by the sum of the importance measures of all metabolites in each pathway [29].

### 2.5. Statistical Analysis

All clinical parameters were computed with Excel (Microsoft 7) and presented as mean ± SD. The comparison among the data was made using one-way repeated measures ANOVA. Further comparisons were made by using paired-sample *t*-test. The SPSS/PC computer program (SPSS, Chicago, IL, USA) was used to perform all statistical analyses. Statistical significance was set at *p* ≤ 0.05. 

The metabolic profile of serum samples from controls and T2DM patients at different stages were analyzed by NMR. The ^1^H-NMR Carr–Purcell–Meiboom–Gill (CPMG) spectra were processed using Topspin 3.5 and Amix 3.9.13 (Bruker, Biospin, Italy), both for simultaneous visual inspection and the successive bucketing process. The full NMR spectra (in the range 9.0–0.5 ppm) were segmented in fixed rectangular buckets of 0.04 ppm width and successively integrated. The spectral region between 5.10 and 4.7 ppm was discarded because of the residual peak of water. The total sum normalization was applied to minimize small differences due to sample concentration and/or experimental conditions among samples. The data set (bucket table) resulted in a matrix, made of 204 variables, corresponding to the bucketed ^1^H-NMR spectra values (in columns), measured for each sample (in rows). Multivariate statistical analysis was performed using MetaboAnalyst software [28]. Unsupervised principal component analysis (PCA), and partial least squares/supervised orthogonal partial least squares discriminant analysis (PLSDA and OPLSDA, respectively) were applied to examine the intrinsic variation in the data, and also to screen out potential biomarkers [30]. In particular, OPLSDA analysis focuses the predictive information in one component, so that the first OPLS component shows the between-class difference. The remaining systematic information is transferred in higher components, thus facilitating interpretation. Two parameters, R^2^ and Q^2^, describe the goodness of the statistical models. The former (R^2^) explains the total variations in the data, whereas the latter (Q^2^, calculated via 10-fold cross-validation, CV) provides an estimate of the predictive ability of the models [31]. By ^1^H-NMR spectroscopy, metabolites of interest were quantified by analyzing the integrals of selected distinctive unbiased NMR signals [32,33,34]. Results, represented as mean intensities and standard deviation of the selected NMR signals, were validated by one-way ANOVA with Tukey’s honestly significant difference (HSD) post-hoc test. To better visualize data, a heatmap was performed on metabolites and samples, using Euclidean for distance measure and Ward for the clustering algorithm. Then, to identify the potential biomarkers associated with T2DM disease in patients with complications (T2DM-C), the receiver operating characteristic (ROC) curve was applied, using the Biomarker Analysis module of the MetaboAnalyst software. Multivariate ROC curve exploratory analysis was used to identify the promising biomarkers with high sensitivity and high specificity. The ROC curves were generated using Monte–Carlo cross validation (MCCV) algorithm and linear Support Vector Machines (SVM) clustering to evaluate the feature importance of the selected metabolites [35]. Both univariate and multivariate statistical analyses were performed using MetaboAnalyst software [28].

## 3. Results

### 3.1. Clinical Characteristics

Three different selected groups of subjects were included for this investigation: non-diabetic subjects, referred as the control group (CG) and two different groups of T2DM patients, with or without complications and insulin treatment. Patients with complications on insulin monotherapy were indicated as T2DM-C; patients without complications and treatments were indicated as T2DM-NC. Clinical characteristics of CG and T2DM groups are summarized in Table 1 and Table 3. There was no significant difference in age (range 60–68 years) and BMI between groups. All subjects were overweight as confirmed by BMI values exceeding 25, with prevalence in android obesity as indicated by the waist to hip ratio (WHR). Fasting plasma glucose level was significantly higher in both T2DM groups compared to CG and significantly different (*p* < 0.05) between T2DM groups. In both T2DM groups, HbA1c and HOMA were higher than in CG, especially in T2DM-C (*p* < 0.001 and *p* < 0.005, respectively, T2DM-C vs. CG). It is important to note that insulin treatment holds plasma insulin levels to values not significantly different from that of other groups, so that metabolic changes we measured in our study cannot be related to defects in insulin secretion.

PAI-1, azotemia and creatinine were significantly increased in T2DM-C. No significant changes in IGF1, β-galactosidase, and α-fucosidase levels were observed between groups. All patients were hyperlipidemic, with no significant differences in their plasma total cholesterol, low-density lipoprotein (LDL), HDL, ApoB, and TAG levels. Ferritin level was significantly higher in T2DM-NC compared to CG (Table 3).

### 3.2. ^1^H-NMR Analysis of Serum Samples

Typical ^1^H-NMR CMPG (600 MHz) spectra of serum, obtained from CG and T2DM samples are reported in Figure 1. Resonance assignments were performed according to the literature [36] and further confirmed by 2D NMR spectra. A complex pattern of signals ascribable to aromatic molecules (1), sugar moieties, and aliphatic metabolites (2) was shown (Figure 1), with some of the identified metabolites reported for each different group. Although the NMR spectra appeared similar among different serum samples, there were striking differences in peak intensities for the different groups. The ^1^H-NMR spectra were dominated by high-intensity signals of sugars (α and β glucose) and some high molecular weight metabolites, such as lipoproteins (very low and low density lipoprotein, VLDL/LDL). As expected, a significant increase in sugar content was observed in T2DM patients (with or without complications) compared to the CG. Small molecules, as branched-chain amino acids (leucine, isoleucine, valine), aliphatic and aromatic amino acids (alanine, arginine, glutamine, glutamate, methionine, glycine, phenylalanine, tyrosine), organic acids (lactate, formate, pyruvate, citrate, acetate, acetoacetate, β-hydroxybutyrate), osmolytes (choline, TMA-N-oxide), and others, including methyl-histidine, *N*-acetyl-glycoproteins, dimethylamine (DMA), trimethylamine (TMA) and creatinine were also identified [27].

### 3.3. Multivariate Analysis of NMR Data

Multivariate data analysis was applied to the NMR spectra for reducing the complexity and the volume of the data. After the pre-processing of the NMR spectra, including the bucketing process and total sum normalization to minimize small differences due to sample concentration and/or experimental conditions among samples [36], both unsupervised (PCA) and supervised (PLSDA, OPLSDA) multivariate statistical methods were applied. As a first attempt, PCA analysis was conducted to display natural groupings of samples without imposing any preconception about class membership, allowing the general trend display and the identification of potential outliers among samples. Supervised statistical methods, such as PLSDA and OPLSDA analyses, were also applied in order to search for discriminating features and potential biomarkers, responsible for the separation between groups. In particular, pairwise multivariate (PCA, PLSDA, OPLSDA) analyses were obtained comparing the CG vs. T2DM-NC (Figure 2) and the CG vs. T2DM-C groups (Figure 3). Moreover, T2DM patients were also compared with each other (T2DM-C vs. T2DM-NC, Figure 4). The PCA score plot obtained for CG and T2DM-NC (Figure 2A) revealed a good separation between the two groups, especially along the first principal component PC1 (PC1 and PC2 accounted for 51.3% and 23% of the total variance, respectively). A clear separation between the two groups was observed also in the corresponding PLSDA (Appendix A in Appendix A) and OPLSDA score plots (Figure 2B). The PLSDA model was obtained, with the first two components explaining 50% and 23.5% of the total variance, R^2^ = 0.86 and Q^2^ = 0.78, and the corresponding OPLSDA score plot was obtained with the first predictive and orthogonal components, accounting for 17.8% and 53% of the total variance, respectively. The variables (bucket reduced NMR signals) responsible for the CG and T2DM-NC separation were observed in the corresponding VIP plot (*p* < 0.05) (Appendix A in Appendix A). In addition to a relative higher α and β-glucose content that characterized T2DM-NC patients with respect to CG, a relative decrease of alanine, creatine/creatinine, glutamine, glutamate, leucine, lysine, methionine, *N*-acetylglycoproteins, phenylalanine, and tyrosine was observed in T2DM-NC patients with respect to the CG (Figure 2C). Moreover, the quantitative variation in discriminating metabolites among the observed groups was calculated by the integration of specific signals, identified by the NMR-based multivariate analysis (Table 4). Results, reported as the mean and standard deviation of integrals for each group, were validated by one-way ANOVA with the HSD post-hoc test. The level of statistical significance was at least at *p*-values < 0.05 with 95% confidence level. PCA, PLSDA, and OPLSDA analyses were also performed for CG and T2DM-C patients (Figure 3). PCA analysis showed an even better separation than that observed comparing CG and T2DM-NC, between CG and T2DM-C patients (PC1 and PC2 accounted for 73.7% and 10.5% of the total variance, respectively) (Figure 3A). This result appeared more evident when PLSDA (with the first two components explaining 73.6% and 9.3% of the total variance, R^2^ = 0.85 and Q^2^ = 0.72) and OPLSDA (with the first predictive and orthogonal components accounting for 22.9% and 48.6%) analyses were applied to the data (Appendix A in Appendix A and Figure 3B, respectively). Together with the expected higher level of sugars, several metabolites, such as alanine, carnitine, citrate, creatine, glutamate, glutamine, isoleucine, leucine, lactate, lysine, methionine, N-acetyl-glycoproteins, phenylalanine, tyrosine and valine were also quantified and statistically validated as significantly reduced in T2DM-C patients compared to the CG (Figure 3C and Table 4). 

Finally, T2DM patients were also compared with each other (T2DM-C vs. T2DM-NC). The statistical models obtained appeared well descriptive but weakly predictive, showing a partial overlap of samples (Figure 4). In fact, PCA model (Figure 4A) was built with PC1 and PC2 accounting for 35.8% and 25.2% of the total variance, while in the PLSDA score plot (Appendix A in Appendix A) the first two components explained 33.5% and 24.5% of the total variance, respectively (R^2^ = 0.46 and Q^2^ = −0.27); the OPLSDA score plot was obtained with the first predictive and orthogonal components accounting for 13.9% and 57.9% (Figure 4B). Metabolites showing a significant variation between the two groups were carnitine, isoleucine, leucine, lactate, lysine, *N*-acetylglycoproteins, tyrosine, and valine (Figure 4C and Table 4).

The metabolites listed in Table 4 were also used as input variables in order to investigate the distance of any type of samples, obtaining a heatmap (Figure 5). Moreover, to identify the potential biomarkers associated with T2DM disease, a multivariate ROC curve analysis was also performed on selected metabolites, considering the T2DM-C patients compared to controls (GC) (Figure 6). An overview of all ROC curves (created from six different biomarker models using different number of features) and predictive accuracies with different features are shown in Figure 6A,B, respectively. Considering the first top metabolites (selected frequency >~0.7%), the obtained AUC value was 0.999, with a 0.938–1 confidence interval (CI). Moreover, from the variable importance in projection (VIP) plot (Figure 6C) and prediction of T2DM-C patients and controls using MCCV analysis (Figure 6D), the most discriminating metabolites in descending order of importance were indicated. In addition to glucose, also leucine, glutamate, lysine, valine, and isoleucine resulted the top five important metabolites, based on their frequency selection during cross-validation (Figure 6C). In conclusion, the ROC curves analysis allowed to obtain a general overview of significant alterations of biomarkers in T2DM patients with complications. In particular, BCAAs such as leucine, valine, and isoleucine resulted significant in this analysis.

### 3.4. Metabolic Pathway Analysis

A metabolic pathway analysis using the MeTPA (Metabolomics Pathway Analysis) software [37] was performed in order to identify the most relevant pathways potentially involved in the observed changes of T2DM serum metabolites. Based on the observed quantitative variation for the identified metabolite content found in CG and T2DM patients at different stages, according to the p-value and the impact value (Appendix A in Appendix A), at least six potential target pathways resulted altered in T2DM-NC patients (Figure 7A). These pathways included alanine, aspartate and glutamate metabolism, D-glutamine and D-glutamate metabolism, lysine degradation, aminoacyl-tRNA biosynthesis, and carbohydrate metabolism (starch and sucrose metabolism, glycolysis, or gluconeogenesis). The same metabolic pathways, with a different significance, appear to be compromised in T2DM-C patients with respect to CG (Figure 7B), with the exception of pathways involving carbohydrate metabolism. Finally, when the analysis was conducted by comparing the two diabetic groups, phenylalanine and branched-chain amino acid (BCAA) metabolism resulted perturbed and discriminant of the two diabetic stages (Figure 7C).

## 4. Discussion

T2DM is a metabolic disorder characterized by chronic hyperglycemia due to the resistance of target tissues to the metabolic action of insulin and dysfunction of pancreatic β cells. According to the most recent guidelines, individuals with T2DM are considered at high to very-high cardiovascular risk, depending on the presence or not of established complications [38]. Insulin therapy is used for the treatment of T2DM when the progression of the disease overcomes the effectiveness of oral hypoglycemic drugs [39]. Due to the paucity of metabolic markers of disease progression and the suboptimal performance of risk equations, the characterization of biomarkers that can indicate specific and temporal metabolic and vascular disturbances in the progression of T2DM is an active field of investigation. Metabolomics has gained growing applications in the identification of circulating biomarkers thus contributing to the diagnosis, prognosis and risk estimation of diseases [40]. In the present study, the serum metabolic profiling of control and T2DM groups was investigated to explore the metabolic response to diabetic complications leading to the identification of a metabolic signature that might serve as predictor of disease progression.

To do this, we profiled a well-characterized diabetic cohort of patients who showed hyperglycemia, with increased levels of HbA1c and HOMA index with respect to CG closely matched for age and BMI. Among T2DM patients, retinopathy, nephropathy, and neuropathy were the most frequent observed complications. Analysis of clinical data revealed that a set of few metabolites including PAI-1, azotemia, and creatinine were increased in the T2DM-C group and potentially related to disease complications. Moreover, an increased ferritin level was observed in T2MD-NC patients compared to the CG (Table 3). This was consistent with previous studies that described a functional correlation between PAI-1, ferritin, and type II diabetes [41,42]. These conventional clinical and biochemical parameters clearly distinguished T2DM patients from CG but do not allow the detection of specific alterations useful for the clinical discrimination of T2DM patient cohorts. Application of metabolomics to these datasets has led to a broader evaluation of a large set of serum metabolites associated with each class of patients. As a result of disease complexity [43], several metabolites were identified in T2DM, most of which were involved in amino acid metabolism (Figure 7D). This result is in agreement with previously reported metabolomic studies highlighting the association of serum amino acids alteration with metabolic disorders, and T2DM [44,45,46,47]. 

More in detail, MetPA software analysis of NMR data showed a significant perturbation of alanine, glutamine and glutamate metabolism in both T2DM groups (T2DM-C and T2DM-NC) of patients (Figure 5, Appendix A). Multivariate analyses pointed out a decreased serum level of these amino acids in both groups of diabetic (T2DM-C and T2DM-NC) with respect to control patients. This result confirms previous studies that identified alanine and glutamine as consistently associated with an increased risk of developing T2DM [36,48]. Alanine, glutamine, and glutamate are gluconeogenic amino acids precursors for glucose synthesis. Therefore, decreased serum level of these amino acids could be associated to a major hepatic utilization for glucose synthesis in agreement with previous reports of increased gluconeogenesis from amino acids in T2DM patients [49,50,51]. Indeed, our metabolic pathway analysis clearly suggests the involvement of the gluconeogenesis pathway, particularly at the earliest T2DM stage (Figure 7). Moreover, the reduced amount of serum lactate, measured in T2DM-C compared with T2DM-NC patients, could be seen as a consequence of the higher alanine utilization for glucose synthesis. In this case, pyruvate is shunted towards alanine and away from lactate synthesis.

The multivariate analysis performed on serum metabolic profiles also showed and allowed to quantify significant differences in the levels of isoleucine, lysine, tyrosine, and valine in T2DM-C with respect to CG patients and between the two diabetic groups (Table 4). Leucine, isoleucine, and valine are essential amino acids also known as BCAAs. Indeed, MetPA analysis highlighted a compromised BCAA metabolism as a discriminant of the two diabetes stages (Figure 7C). Thus, the strong decrease in BCAA serum level observed in the advanced with respect to T2DM early stage could be considered as a time-dependent perturbation of BCAA metabolism. Moreover, ROC curve analysis validated the clinical relevance of BCAAs as potential biomarkers in the diagnosis of T2DM complications (Figure 6). 

In recent years, BCAA metabolism was found to be significantly altered in patients with several diseases including diabetes [52]. Both human and animal studies demonstrated a functional correlation between BCAA metabolism and a higher susceptibility to IR in T2DM patients [53,54,55,56,57]. Here, we showed lower levels of BCAA in T2DM patients compared to CG, raising questions regarding the mechanisms regulating BCAA metabolism in our cohort. In this context, we speculate that IR levels and disease complications may have a major impact on this result, as corroborated by other studies that utilize functional and NMR approaches. For instance, lower plasma levels, with respect to healthy controls, of BCAAs in T2DM patients were indicated as a first sign of kidney dysfunction [56]. Further, the decreased serum levels of BCAAs that we observed in T2DM are in line with the results of Shin et al. [57] who demonstrated that alteration in hypothalamic insulin signaling can decrease plasma BCAA levels by inducing the hepatic activity of branched-chain α-keto acid dehydrogenase, a rate-limiting enzyme in the BCAA degradation pathway. Overall, this indication well correlated with the HOMA-IR values that we measured in the diabetic patients with respect to CG, with the highest value of IR measured in the complicated T2DM patients (Table 3). Moreover, a higher BCAA level has been found in retinal Müller cells from diabetic compared to euglycemic rats [58]. This study demonstrated that BCAAs competitively inhibited glutamate transamination and induced glutamate excitotoxicity and neuronal cell death, providing an early sign of retinopathy. This data well correlated with the high occurrence (more than 90% of subjects) of retinopathy that we measured among T2DM-C patients (Table 1). Moreover, increased muscle BCAA oxidation has been reported to improve muscle glucose uptake in metabolic syndrome by enhancing the recycling of glucose via the glucose-alanine cycle [59].

Serum levels of lysine decreased from control subjects to T2DM patients at an early stage and to T2DM complicated patients. Interestingly, recent work reported a therapeutic effect of lysine administration in T2DM patients to counteract the production of glycated lysozyme [60]. Protein glycation is a well-established process associated with long-term hyperglycemia characterizing many pathophysiological conditions such as cancer, inflammation, metabolic dysfunctions and aging [61].

We also found that tyrosine serum level significantly decreased in both groups of diabetic patients with respect to CG and also differentiated T2DM-C from T2DM-NC patients (Table 4). It is important to note that a previous work, conducted on rat liver, demonstrated that insulin was capable to increase the activity of the tyrosine aminotransferase enzyme. Thus, by activating the transamination of tyrosine to p-hydroxyphenylpyruvate the serum level of tyrosine was reduced [62]. On the basis of these data, the high fasting insulin level measured in diabetic subjects (Table 3) could be linked to the decreased tyrosine serum level in diabetic patients. Importantly, it has been also reported that circulating tyrosine level could be correlated with the risk of developing various major complications of diabetes [56]. Moreover, other studies described an association of a low tyrosine level with the impairment of kidney function, which itself could predict future microvascular events [63,64]. Tyrosine is also linked to catecholamine synthesis, a mechanism that may be involved in the development of T2DM complications [65]. Increasing evidence suggested a correlation among tyrosine levels, obesity, and insulin concentration in both diabetic and non-diabetic subjects [65]. Moreover, tyrosine has been identified as the only metabolite significantly associated with HOMA in obesity-independent models [66,67]. 

The serum level of patients in the advanced T2DM stage was also characterized by a lower level of carnitine with respect to T2DM-NC. L-carnitine, primarily synthesized in the liver and kidneys from lysine and methionine, has an essential role in the transfer of activated long-chain fatty acids into the mitochondria where β-oxidation takes place [68]. Considering that impaired fatty acid transport inside mitochondria can result in triglyceride cytosolic accumulation in and IR [69], we can consider the reduced serum carnitine level measured in complicated T2DM patients closely associated to the highest HOMA-IR of these subjects.

## 5. Conclusions

Results from the present study identified a set of metabolites discriminating control from T2DM groups. We also demonstrated that specific metabolic alterations characterize T2DM groups at the systemic level with main differences in amino acids pathways. If this result reflects a way to decrease the pathological state by activating compensatory mechanisms or is a mere consequence of deranged metabolic pathways is yet to be understood. In this context, the metabolism of gluconeogenic amino acids and BCAAs may have a role. On the basis of these results, considering that BCAAs are essential amino acids introduced with the diet, changes in dietary habit, not only as carbohydrate intake, have to be taken into account for the prevention of T2DM complications. Finally, if this metabolic change represents the pathological basis for the onset of complications increasing, for instance, the risk of developing cancer should be also considered and evaluated. Our data will represent the basis for future studies involved patients affected by tumors related to metabolic dysfunctions such as hepatocellular carcinoma.

## Figures and Tables

**Figure 1 jcm-08-00720-f001:**
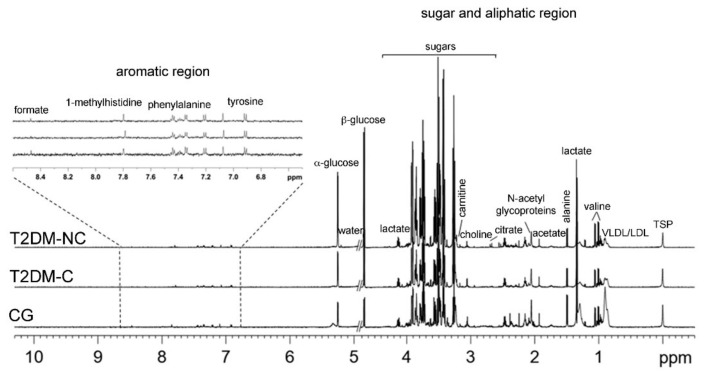
Representative ^1^H CPMG (Carr–Purcell–Meiboom–Gill) NMR spectra of serum isolated from different groups of diabetic (T2DM-NC, T2DM-C) and control (CG) patients. Aromatic, and sugar and aliphatic regions with some identified metabolites were visualized.

**Figure 2 jcm-08-00720-f002:**
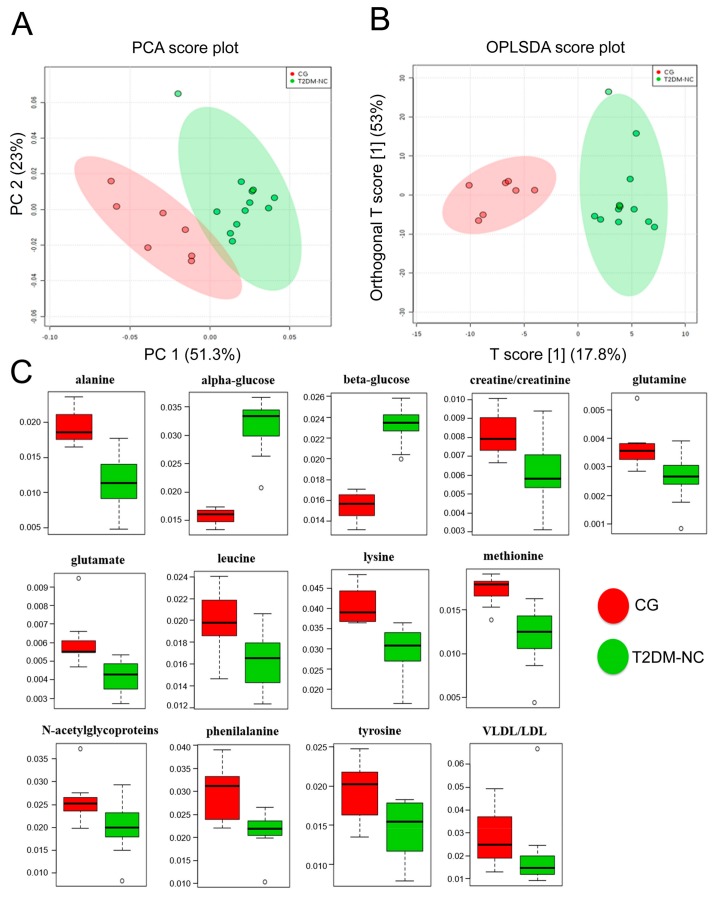
PCA and OPLSDA score scatter plots distinguishing between CG and T2DM-NC. (**A**) PCA (PC1/PC2, 51.3% and 23% of the total variance, respectively); (**B**) OPLSDA (obtained with the first predictive and orthogonal components, accounting for 17.8% and 53% of the total variance, respectively) analyses of NMR data; (**C**) metabolites significantly changed between T2DM-NC and CG patients. PCA = principal component analysis; OPLSDA = orthogonal partial least squares discriminant analysis.

**Figure 3 jcm-08-00720-f003:**
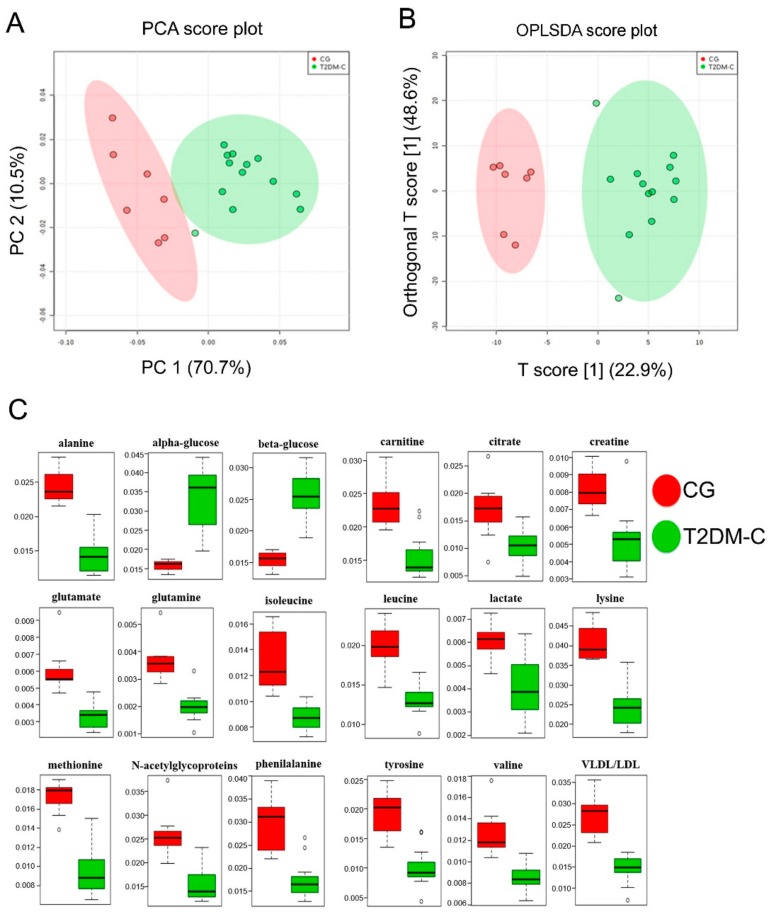
PCA and OPLS-DA score scatter plots distinguishing between CG and T2DM-C. (**A**) PCA (PC1/PC2, 73.7% and 10.5% of the total variance, respectively); (**B**) OPLSDA (obtained with the first predictive and orthogonal components, accounting for 22.9% and 48.6% of the total variance, respectively) analyses of NMR data; (**C**) metabolites significantly changed between T2DM-C and CG patients.

**Figure 4 jcm-08-00720-f004:**
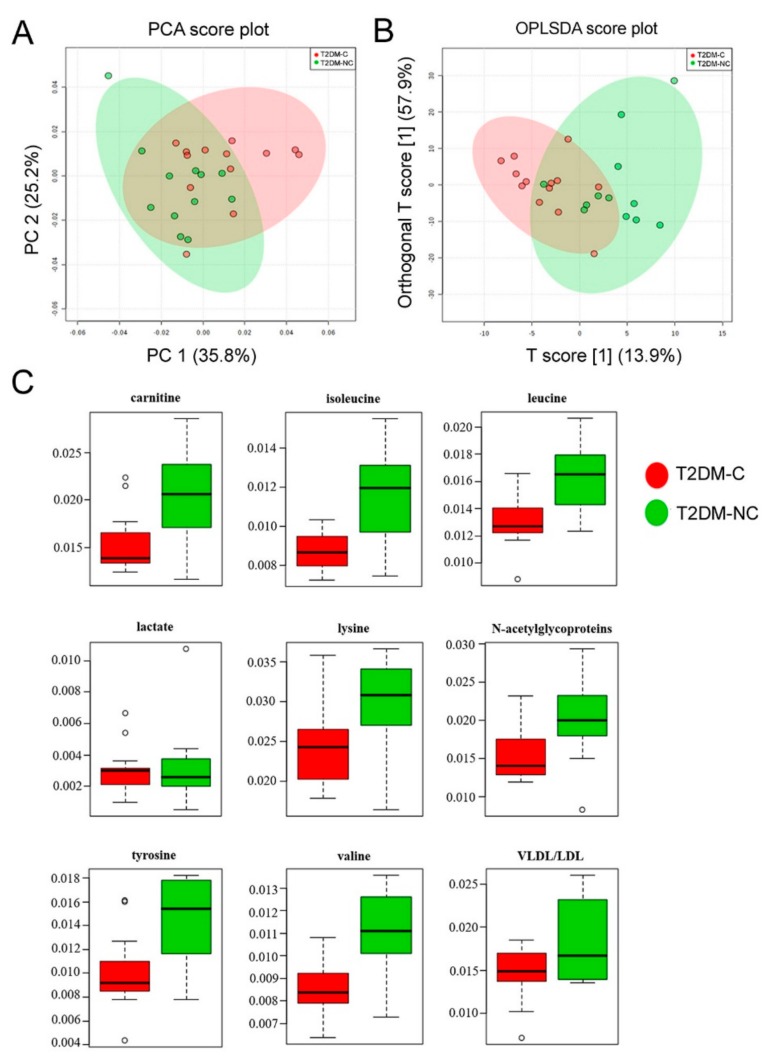
PCA and OPLS-DA score scatter plots distinguishing between T2DM-NC and T2DM-NC. (**A**) PCA (PC1/PC2, 35.8% and 25.2% of the total variance, respectively); (**B**) OPLSDA (obtained with the first predictive and orthogonal components, accounting for 13.9% and 57.9% of the total variance, respectively) analyses of NMR data; (**C**) metabolites significantly changed between T2DM-C and T2DM-NC patients.

**Figure 5 jcm-08-00720-f005:**
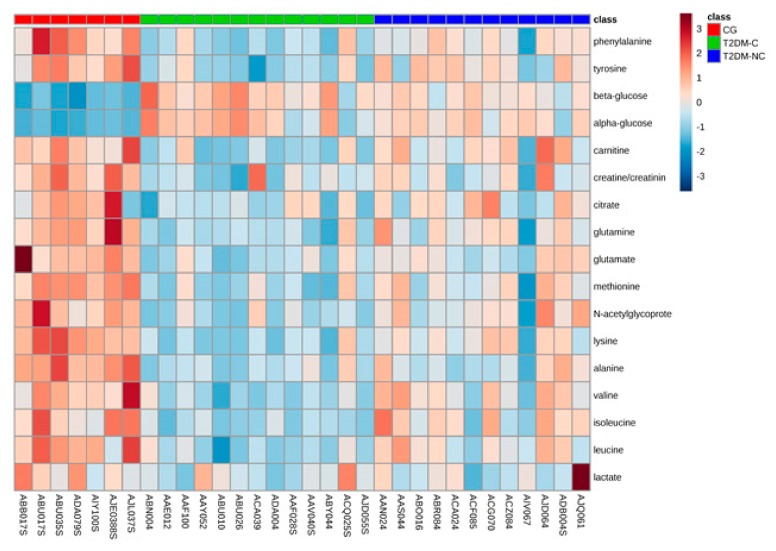
Heatmap visualization of group averages based on 17 biomarkers. Heatmap visualization based on 17 metabolites. Rows: biomarkers; columns: samples. Red: GC; green and blue: T2DM-C and T2DM-NC patients, respectively. Color key indicates metabolite values: dark blue: lowest; dark red: highest.

**Figure 6 jcm-08-00720-f006:**
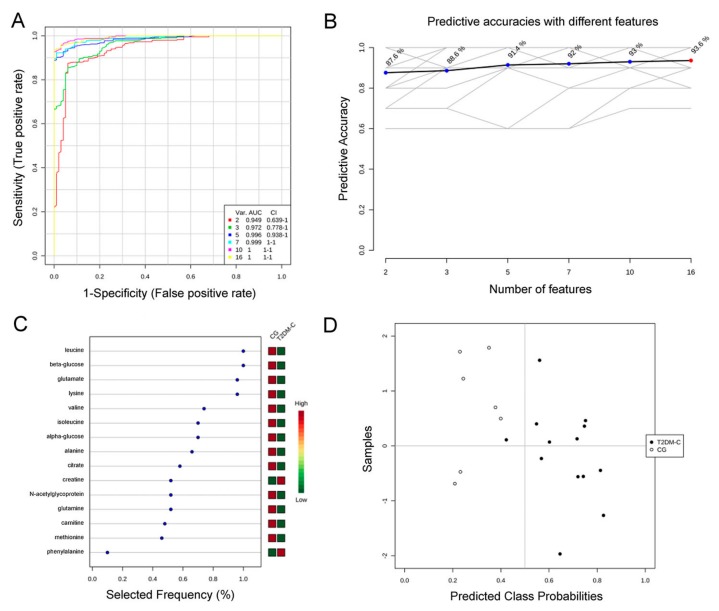
Comparison of variables based on ROC curve. (**A**) Multivariate receiver operating characteristic (ROC) analysis, showing the feature numbers the AUCs and the confidence intervals of the six models; (**B**) predictive accuracies with different features based on the ROC curves; (**C**) percentage selected frequency of metabolites based on ROC curves, with the variable importance in projection (VIP) plot indicating the most discriminating metabolite in descending order of importance; (**D**) prediction of T2DM-C patients and controls using Monte–Carlo cross validation (MCCV) analysis.

**Figure 7 jcm-08-00720-f007:**
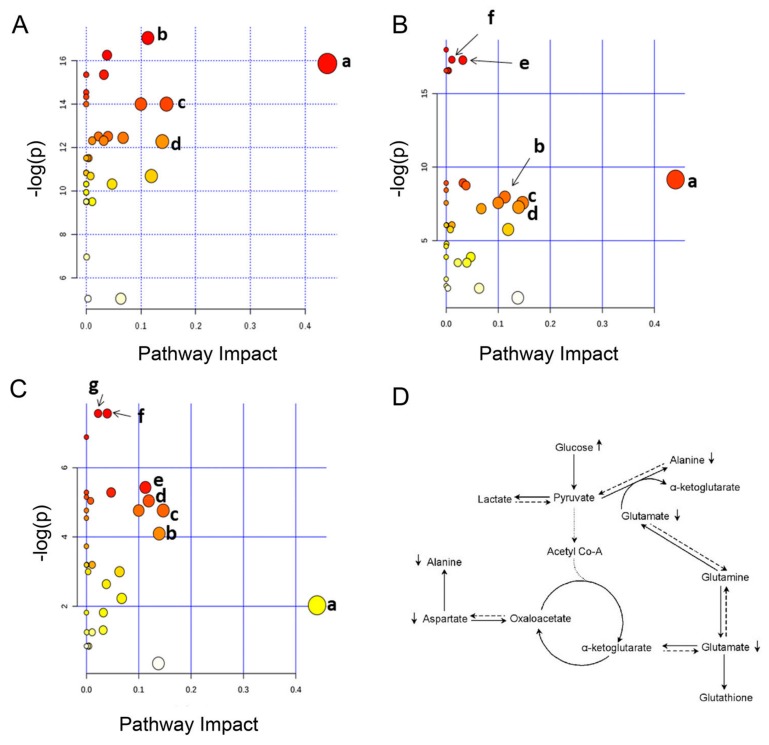
Summary of metabolic analysis conducted by MetPA. (**A**) CG vs. T2DM-C. (**a**) Alanine, aspartate, and glutamate metabolism; (**b**) aminoacyl-tRNA biosynthesis; (**c**) lysine degradation; (**d**) d-glutamine and d-glutamate metabolism. (**B**) CG vs. T2DM-NC. (**a**) Alanine, aspartate and glutamate metabolism; (**b**) aminoacyl-tRNA biosynthesis; (**c**) lysine degradation; (**d**) d-glutamine and d-glutamate metabolism; (**e**) starch and sucrose metabolism; (**f**) glycolysis or gluconeogenesis. (**C**) T2DM-NC vs. T2DM-C. (**a**) Alanine, aspartate and glutamate metabolism; (**b**) d-glutamine and d-glutamate metabolism; (**c**) lysine degradation; (**d**) phenylalanine metabolism; (**e**) aminoacyl-tRNA biosynthesis; (**f**) valine, leucine and isoleucine biosynthesis; (**g**) valine, leucine and isoleucine degradation. (**D**) Schematic diagram of the metabolic pathways supposed to be altered in T2DM detected by ^1^H NMR analysis. Arrows (↑ ↓) represented the increase or decrease of metabolites in the serum.

**Table 1 jcm-08-00720-t001:** Number and type of complications in T2DM-C patients.

	Number
**Median Age (Range)**	64 (55–72)
**Gender**	
Male	7/13
Female	6/13
**Complications**	
Neuropathy	9/13
Nephropathy	7/13
Retinopathy	12/13
Chronic renal failure	4/13
Lower limb arteriopathy	5/13
Mace	5/13

**Table 2 jcm-08-00720-t002:** Drug treatment prescribed to T2DM patients.

	T2DM-NC	T2DM-C
ACE inhibitors	1	8
Diuretics	-	5
Vasodilators	-	4
Beta blockers	-	4
Antiarrhythmic drugs	-	2
Calcium channel blockers	-	3
Statins	1	1
Antiplatelet drugs	-	5
NSAIDs	-	7
Proton-pump inhibitors	1	2
CNS agents	2	2

Abbreviations: ACE = angiotensin-converting-enzyme; Abbreviations: NSAIDs = nonsteroidal anti-inflammatory drugs; CNS = central nervous system.

**Table 3 jcm-08-00720-t003:** Clinical information for control group (CG) and type 2 diabetes mellitus (T2DM) patients (T2DM-NC and T2DM-C) enrolled in the study. (T2DM-C: patients with complications on insulin monotherapy; T2DM-NC: without complications).

	CG	T2DM-NC	T2DM-C
Number of subjects (*n*)	7	13	13
Male gender (*n*, %)	4, 57.1%	8, 61.5%	7, 53.8%
Age (years)	63 ± 2	64 ± 3	64 ± 4
BMI (kg/m^2^)	26.94 ± 3.05	28.78 ± 4.05	29.31 ± 3.82
WHR	0.90 ± 0.07	0.94 ± 0.06	0.94 ± 0.06
Fasting glucose (mg/dL)	93 ± 6.2	165 ± 59.4 ***^,b^	247 ± 65.0 ***^,a^
HbA1c (%)	5.68 ± 0.38	6.92 ± 0.90 ***^,b^	8.87 ± 1.73 ***^,a^
Fasting insulin (mU/L)	5.2 ± 1.6	9.2 ± 7.8	8.5 ± 4
hsCRP (mg/L)	2.4 ± 2.1	3.5 ± 3.1	3.9 ± 2.8
PAI-1 (ng/mL)	18.9 ± 5.7	24.7 ± 11	16 ± 8*
Creatinine (mg/dL)	0.81 ± 0.17	0.91 ± 0.19	1.33 ± 0.70 *
Azotemia (mg/dL)	38 ± 7.9	38 ± 9.7 ^b^	54 ± 30 **^,a^
Ferritin (ng/mL)	111 ± 71	236 ± 177 **	140 ± 113
Total cholesterol (mg/dL)	237 ± 25	230 ± 36	215 ± 43
HDL cholesterol (mg/dL)	59 ± 11	58 ± 15	51 ± 14
LDL cholesterol (mg/dL)	141 ± 24	136 ± 36	116 ± 34
Triglycerides (mg/dL)	106 ± 46	142 ± 128	160 ± 95
ApoB (mg/dL)	109 ± 2	112.8 ± 30	105.1 ± 30
IGF1 (ng/mL)	32.7 ± 6.5	38.5 ± 9.7	32.9 ± 5.8
β-Galactosidase (nM/ml/h)	5.47 ± 1.73	3.77 ± 3.11	6.65 ± 3.87
α-Fucosidase (nM/ml/h)	374 ± 179	328 ± 208	389 ± 219
HOMA-IR	1.2 ± 0.4	2.9 ± 2.0 *	4.3 ± 1.9 ***
eGRF (ml/min)	89.7 ± 28.1	82.8 ± 14.7	60.5 ± 25.3 *
Disease duration (years)	n/a	n/a	22 ± 12

Values are given as the mean ± SD. *p* values were obtained by ANOVA and post-Tukey. *, ** and *** refer to a *p*-value < 0.05, 0.005 and < 0.001, respectively, for the type 2 diabetes mellitus (T2DM) classes versus control group (CG). Different letters represent significant differences among different T2DM groups. T2DM-C = T2DM patients with complications and insulin treatment; T2DM-NC = T2DM patients without complications and treatment; BMI = body mass index; WHR = waist to hip ratio; PAI-1 = plasminogen activator inhibitor-1; IGF1 = Insulin growth factor; HOMA-IR = Homeostatic Model Assessment for Insulin Resistance; eGRF = estimated glomerular filtration rate; ApoB = apolipoprotein B; hsCRP = highly sensitive C-reactive protein.

**Table 4 jcm-08-00720-t004:** Quantitative comparison of metabolites found in the serum of T2DM and CG patients.

Metabolite	Chemical Shift (ppm)	Integral in CG Group ^a^ (Mean ± SD) × 10^−2^	Integral in T2DM-NC ^a^ Group (Mean ± SD) × 10^−2^	Integral in T2DM-C ^a^ Group (Mean ± SD) × 10^−2^	*p*-Value	Tukey’s HSD Adjusted *p*-Value (FDR) Cutoff: 0.05 ^b^
Alanine	1.49 (d)	1.95 ± 0.27	1.14 ± 0.38 (↓)	0.96 ± 0.25 (↓)	6.89 × 10^−7^	T2DM-C/CG; T2DM-NC/CG
α-glucose	5.25 (d)	1.57 ± 0.14	3.16 ± 0.45 (↑)	3.37 ± 0.77 (↑)	5.35 × 10^−7^	T2DM-C/CG; T2DM-NC/CG
β-glucose	4.66 (d)	1.54 ± 0.14	2.33 ± 0.18 (↑)	2.56 ± 0.33 (↑)	6.26 × 10^−9^	T2DM-C/CG; T2DM-NC/CG
Carnitine	3.22 (s)	2.35 ± 0.4	2.04 ± 0.45 (↓)	1.54 ± 0.33 (↓)	3.38 × 10^−4^	T2DM-C/CG; T2DM-NC/T2DM-C
Citrate	2.65 (d)-2.56(d)	0.17 ± 0.006	0.14 ± 0.04 (↓)	0.11 ± 0.035 (↓)	1.2 × 10^−2^	T2DM-C/CG
Creatine/creatinine	3.05 (s)	0.82 ± 0.13	0.6 ± 0.16 (↓)	0.53 ± 0.17 (↓)	2.1 × 10^−3^	T2DM-C/CG; T2DM-NC/CG
Glutamate	2.38 (m)	0.61 ± 0.16	0.41 ± 0.08 (↓)	0.34 ± 0.08 (↓)	2.35 × 10^−5^	T2DM-C/CG; T2DM-NC/CG
Glutamine	2.45 (m)	0.37 ± 0.008	0.26 ± 0.07 (↓)	0.2 ± 0.05 (↓)	5.34 × 10^−5^	T2DM-C/CG; T2DM-NC/CG
Isoleucine	1.02 (d)	1.32 ± 0.25	1.15 ± 0.24 (↓)	0.88 ± 0.1 (↓)	1.15 × 10^−4^	T2DM-C/CG; T2DM-NC/T2DM-C
Leucine	0.98 (d)	1.99 ± 0.31	1.64 ± 0.24 (↓)	1.32 ± 0.2 (↓)	9.48 × 10^−6^	T2DM-C/CG; T2DM-NC/CG; T2DM-NC/T2DM-C
Lactate	1.33 (d)	6.05 ± 0.82	5.4 ± 1.6 (↓)	4.1 ± 1.4 (↓)	1.18 × 10^−2^	T2DM-C/CG; T2DM-NC/T2DM-C
Lysine	1.74 (m)	0.41 ± 0.005	0.29 ± 0.05 (↓)	0.24 ± 0.05 (↓)	6.20 × 10^−7^	T2DM-C/CG; T2DM-NC/CG; T2DM-NC/T2DM-C
Methionine	2.14 (m)	1.72 ± 0.19	1.21 ± 0.33 (↓)	0.96 ± 0.26 (↓)	1.17 × 10^−5^	T2DM-C/CG; T2DM-NC/CG
*N*-acetylglycoproteins	2.05 (m)	2.61 ± 0.55	2.02 ± 0.55 (↓)	1.53 ± 0.35 (↓)	0.2 × 10^−3^	T2DM-C/CG; T2DM-NC/CG; T2DM-NC/T2DM-C
Phenylalanine	7.34 (d)	0.29 ± 0.006	0.21 ± 0.04 (↓)	0.17 ± 0.04 (↓)	2.54 × 10^−5^	T2DM-C/CG; T2DM-NC/CG
Tyrosine	7.18 (m)	1.92 ± 0.004	0.14 ± 0.04 (↓)	0.1 ± 0.03 (↓)	4.09 × 10^−5^	T2DM-C/CG; T2DM-NC/CG; T2DM-NC/T2DM-C
Valine	1.04 (d)	1.28 ± 0.24	1.10 ± 0.19 (↓)	0.86 ± 0.12 (↓)	7.35 × 10^−5^	T2DM-C/CG; T2DM-NC/T2DM-C

The arrows (↑/↓) were used to show the metabolite levels increase/decreased compared with CG. ^a^ Mean and relative standard deviation refers to the relative integrals of metabolites, determined from serum 1D ^1^H-NMR spectra of each group. The chemical shifts used for calculating integrals were reported in the second column. ^b^ An FDR (False Discovery Rate) adjusted *p*-value of 0.05 was obtained from analysis of variance (one way-ANOVA) with Tukey’s honestly significant difference (HSD) post hoc test. The levels of statistical significance were at least at *p*-values < 0.05 with 95% confidence level. Letters in parentheses indicate the peak multiplicities: s, singlet; d, doublet; m, multiplet).

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
