# Peer review of "NMR-Based Metabolomic Approach Tracks Potential Serum Biomarkers of Disease Progression in Patients with Type 2 Diabetes Mellitus"

_jcm, 2019, doi:10.3390/jcm8050720_

Round 1
Reviewer 1 Report
The article by Del Coco et al, has applied applied a 1H-Nuclear Magnetic Resonance (1H-NMR)-based metabolomic approach, coupled with multivariate data analysis, to identify serum metabolite profiles associated with T2DM development and progression. They have compared serum metabolome of non-diabetic subjects, treatment-naïve non-complicated T2DM patients, and T2DM patients with complications in insulin monotherapy. They have obtained a significant reduction of alanine, glutamine, glutamate, leucine, lysine, methionine, tyrosine and phenylalanine in T2DM patients with respect to non-diabetic subjects. Moreover, isoleucine, leucine, lysine, tyrosine and valine levels distinguished complicated patients from patients without complications. Overall, the metabolic pathway analysis suggested that branched-chain amino acids (BCAA) metabolism is significantly compromised in T2DM patients with complications, while perturbation in the metabolism of gluconeogenic amino acids other than BCAA characterizes both early and advanced T2DM stages. They conclude that they have identified a metabolic serum signature associated with T2DM stages. These data could be integrated with clinical characteristics to build a composite T2DM\complications risk score to be validated in a prospective cohort.
I have the following comments.
- Authors have described a group of patients with or without complications, but there is a different grade of complications, suggesting that the groups can have a big difference for example in nephropathy evolution, or retinopathy.
- The physiological and clinical implication of the data should be explained widely.
- Mitochondrial parameters should be evaluated in order to know the physiological mechanism and metabolism.
- Time of diabetes evolution should be shown.
- Cardiovascular parameters should be shown.
- Treatments should be shown.
- Some figures are very difficult to understand or even to read.
Author Response
Reviewer 1
Comments and Suggestions for Authors
The article by Del Coco et al, has applied a 1H-Nuclear Magnetic Resonance (1H-NMR)-based metabolomic approach, coupled with multivariate data analysis, to identify serum metabolite profiles associated with T2DM development and progression. They have compared serum metabolome of non-diabetic subjects, treatment-naïve non-complicated T2DM patients, and T2DM patients with complications in insulin monotherapy. They have obtained a significant reduction of alanine, glutamine, glutamate, leucine, lysine, methionine, tyrosine and phenylalanine in T2DM patients with respect to non-diabetic subjects. Moreover, isoleucine, leucine, lysine, tyrosine and valine levels distinguished complicated patients from patients without complications. Overall, the metabolic pathway analysis suggested that branched-chain amino acids (BCAA) metabolism is significantly compromised in T2DM patients with complications, while perturbation in the metabolism of gluconeogenic amino acids other than BCAA characterizes both early and advanced T2DM stages. They conclude that they have identified a metabolic serum signature associated with T2DM stages. These data could be integrated with clinical characteristics to build a composite T2DM\complications risk score to be validated in a prospective cohort.
I have the following comments.
- Authors have described a group of patients with or without complications, but there is a different grade of complications, suggesting that the groups can have a big difference for example in nephropathy evolution, or retinopathy. The physiological and clinical implication of the data should be explained widely.
- We are grateful to the reviewer for these constructive comments. We focus our paper on diabetic complications and to perform this task we enrolled healthy subjects, overweigh uncomplicated and complicated T2DM patients. Concerning the clinical data of complicated patients and their relation with metabolomics, we have now added in Table 1, in addition to creatinine levels already reported, values of Glomerular Filtration Rate estimation (eGFR). This may provide a further indication of the clinical characteristics of our cohort in relation to their complications.Outcomes of this study suggest that the application of metabolomics in diabetic patients has the potential to profile uncomplicated and complicated T2DM patients, leading to the identification of a dataset of metabolites with a potential clinical role. This opens to further clinical studies on larger patient cohorts to specifically validate and correlate this signature to T2DM complications.
Physiologically, data highlighted a complex interaction between insulin-resistance, obesity and circulating amino acid levels. We observed a significant BCAA reduction both in uncomplicated and complicated T2DM patients compared to healthy subjects. Both groups of patients were characterized by adequate insulin plasma levels, either endogenous (uncomplicated patients) or exogenous (complicated patients). In this framework, the reduction in the ability of cells to take glucose leads to the activation of an adaptive metabolic reprogramming that is critical for the maintaining of metabolic homeostasis. Under these conditions, this systemic imbalance may promote cellular uptake of BCAA, thus reducing their circulating levels, as also supported by data from animal models (Ref. 57).
- Mitochondrial parameters should be evaluated in order to know the physiological mechanism and metabolism.
- We are grateful to the reviewer for this comment. By analysing serum metabolome we can indirectly extrapolate some data on mitochondrial functionality. To this respect, and in accordance with [Niewczas et al. Kidney International, 2014; Huang et al. Molecular BioSystems, 2013] we found that BCAA and aromatic AA were significantly decreased. These data can indicate an increased mitochondrial AA β-oxidation. Future studies will be focused on a better functional characterization of mitochondrial amino acid metabolism in T2DM progression.
- Time of diabetes evolution should be shown.
- We are grateful to the reviewer for this comment. Data on disease duration were added in Table 1 of the revised version of the manuscript.
- Cardiovascular parameters should be shown.
- We are grateful to the reviewer for this comment, but unfortunately, clinical cardiovascular parameters are not available.
- Treatments should be shown.
- We are grateful to the reviewer for this comment. A new table (Table 3) describing T2DM patients treatments has now been added in the revised version of the manuscript.
- Some figures are very difficult to understand or even to read.
- We are grateful to the reviewer for this comment. Lettering in Figure 6 was now modified at 300 dpi. We also increase the size of all the other figures. Moreover, we also better described Figure 6 panels in the text (pages 12 and 13 lines 335-347)
Reviewer 2 Report
1. The authors must clarify how to identification these metabolites by NMR, what database are used?
2. The correlation between diabetes are BCAA metabolism have been well studied in previous metabolomics studies, the author have to make it clear what is the new discoveries in the current study.
3. In table 3, the author should clarify the results are in relative quantification or absolute quantification.
4. In Figure 2 and 3, please explain why to choose OPLC-DA analysis?
5. In figure 3, please explain why the two group cannot be separated by OPLS-DA?
Author Response
Reviewer 2:
The authors must clarify how to identification these metabolites by NMR, what database are used?Lines 145-147: As already reported in the manuscript, NMR peak assignments were carried out using 2D NMR spectra (1H-1H J-resolved, 1H-1H COSY, 1H-13C HSQC, 1H-13C HMBC) randomly performed on samples and by comparison with published data [Beckonert, O.; Keun, H.C.; Ebbels, T.M.; Bundy, J.; Holmes, E.; Lindon, J.C.; Nicholson, J.K. Metabolic profiling, metabolomic and metabonomic procedures for NMR spectroscopy of urine, plasma, serum and tissue extracts. Nat Protoc 2007,2,2692; Nicholson, J.K.; Foxall, P.J.; Spraul, M.; Farrant, R.D.; Lindon, J.C. 750 MHz 1H and 1H-13C NMR spectroscopy of human blood plasma. Anal Chem 1995,67,793-811]. Any specific database was used in this study.
- The correlation between diabetes are BCAA metabolism have been well studied in previous metabolomics studies, the author have to make it clear what is the new discoveries in the current study.
- We are grateful to the reviewer for this constructive comment. The main purpose and novel aspect of our study was to establish whether the metabolomic approach was able to discriminate between T2DM opposite phenotypes, such as uncomplicated and complicated T2DM patients. We enrolled healthy subjects, overweigh uncomplicated and complicated T2DM patients, in order to track the diabetic progression, rather than to evaluate the contribution of the metabolome on the development of specific complications. Our investigation highlighted a complex interaction between insulin-resistance, obesity and circulating amino acid levels. We observed a significant BCAA reduction both in uncomplicated and complicated T2DM patients compared to healthy subjects. Both groups of patients were characterized by adequate insulin plasma levels, either endogenous (uncomplicated patients) or exogenous (complicated patients). In this framework, we hypothesized that insulin can promote cellular uptake and metabolism of BCAA, thus reducing their circulating levels, hypothesis strongly supported by data from animal models (57).
- In table 3, the author should clarify the results are in relative quantification or absolute quantification.
- Lines 285-286: As already stated in the text, in Table 3, became Table 4 in the modified version of the work, a relative quantification is reported. In fact, mean and relative standard deviation refer to the relative integrals of metabolites, determined from serum 1D 1H-NMR spectra of each group.
- In Figure 2 and 3, please explain why to choose OPLC-DA analysis?
- The OPLS-DA analysis reported in figure 2B and 3B were performed for T2DM-NC and CG and T2DM-C and CG patients, respectively. OPLS-DA analysis is suitable for diagnosing differences between two groups even in the presence of confounding factors. As already reported in the manuscript (Lines 175-178), OPLS-DA analysis focuses the predictive information in one component, so that the first OPLS component shows the between-class difference, thus facilitating interpretation. Nevertheless, the corresponding PLS-DA analyses were reported in Supplementary Information, as Figures S2 and S3, respectively.
- In figure 3, please explain why the two group cannot be separated by OPLS-DA?
- Lines 305-308: as already stated in the text, if the reviewer refers to Figure 4, the OPLS-DA analysis was performed exclusively on patients which were compared with each other (T2DM-C vs T2DM-NC), obtaining a less marked separation between the two groups. Nevertheless, also in this case, a certain degree of separation allowed the identification of metabolites with a significant variation between the two groups (carnitine, isoleucine, leucine, lactate, lysine, N-acetylglycoproteins, tyrosine and valine), reported in Table 3 and Figure 4C.
Round 2
Reviewer 1 Report
no more comments
Reviewer 2 Report
The authors have addressed most of the questions. No further comments.